Subject-specific body segment parameter estimation using 3D photogrammetry with multiple cameras

Peyer Kathrin E. kathrin.peyer@manchester.ac.uk
Morris Mark
Sellers William I.
Faculty of Life Sciences, University of Manchester , Manchester , United Kingdom
Maschner Herbert
Electronic publication date: 2015 Mar 10
Publication date: 2015
Volume: 3
Electronic Location ID: e831
Received 2014 Nov 3; Accepted 2015 Feb 18
Copyright: © 2015 Peyer et al.
Copyright year: 2015
Copyright holder: Peyer et al.
License: This is an open access article distributed under the terms of the Creative Commons Attribution License, which permits unrestricted use, distribution, reproduction and adaptation in any medium and for any purpose provided that it is properly attributed. For attribution, the original author(s), title, publication source (PeerJ) and either DOI or URL of the article must be cited.
License URL: https://creativecommons.org/licenses/by/4.0/

Keywords: Body segment parameters, Photogrammetry, Structure from motion, Subject-specific estimation, Geometric modelling, Biomechanics

Funding: BBSRC BB/K006029/1 Research was funded by the BBSRC (grant number BB/K006029/1). The funders had no role in study design, data collection and analysis, decision to publish, or preparation of the manuscript.

==============================
Inertial properties of body segments, such as mass, centre of mass or moments of inertia, are important parameters when studying movements of the human body. However, these quantities are not directly measurable. Current approaches include using regression models which have limited accuracy: geometric models with lengthy measuring procedures or acquiring and post-processing MRI scans of participants. We propose a geometric methodology based on 3D photogrammetry using multiple cameras to provide subject-specific body segment parameters while minimizing the interaction time with the participants. A low-cost body scanner was built using multiple cameras and 3D point cloud data generated using structure from motion photogrammetric reconstruction algorithms. The point cloud was manually separated into body segments, and convex hulling applied to each segment to produce the required geometric outlines. The accuracy of the method can be adjusted by choosing the number of subdivisions of the body segments. The body segment parameters of six participants (four male and two female) are presented using the proposed method. The multi-camera photogrammetric approach is expected to be particularly suited for studies including populations for which regression models are not available in literature and where other geometric techniques or MRI scanning are not applicable due to time or ethical constraints.

Introduction

Inertial body segment parameters (BSP) such as mass, centre of mass (CoM) or moment of inertia are used in motion analysis in research as well as in clinical settings. Accurate values are essential for techniques such as inverse dynamic analysis to allow the calculation of joint torques based on measured segmental accelerations (Winter, 1979). However, it is not straightforward to measure these quantities from subjects directly. One approach is to use mathematical models of the body segments and rely on anthropometric measurements to determine the dimensions of the modelled segments. This type of methods requires a multitude of anthropometric measurements of the participants and is limited by the accuracy of the mathematical model of the body segments. The first mathematical model suggested by Hanavan in 1964 represented 15 body segments as cylinders and spheres and required 25 anthropometric measurements (Hanavan, 1964). More detailed models presented by Hatze or Yeadon required a total of 95 or 242 measurements, respectively, rendering these methods inefficient for studies with a large number of participants because of the time and discomfort experienced by the participant to acquire all the measurements needed (Hatze, 1980; Yeadon, 1990). Other types of approaches rely on X-ray or MRI based tomography to extract subject-specific BSP from participants. Unlike other methods, CT or MRI scans provide information about internal structures such as tissue composition which should improve the reconstruction accuracy (Martin et al., 1989; Mungiole & Martin, 1990; Pearsall, Reid & Livingston, 1996; Bauer et al., 2007). However, these approaches are also difficult to implement in large-scale studies due to cost and ethical constraints. Alternatively, it is possible to approximate inertial BSP by adjusting previously reported average values or using regression models that require only a very few subject-specific measurements (commonly subject height and weight). Such average values and regression models were derived from cadavers or participants in a number of famous studies, such as the ones by Clauser, Dempster or Zatsiorsky (via de Leva) (Dempster, 1955; Clauser, McConville & Young, 1969; McConville, Clauser & Churchill, 1980; Leva, 1996). However, the reliability of such regression models is rather low, and the models are only applicable to a population similar to the one used to derive the regression equations.

Recently, other methods have been explored to obtain volumetric data of body segments that, in combination with body density assumptions, can provide subject-specific inertial BSP. Sheets, Corazza & Andriacchi (2010) used a laser to scan the body surface of participants and morph a generic model, which contained joint location information, to the scanned surface. Bonnechère et al. (2014) used a Kinect sensor to estimate body segment lengths but not the volumetric data required to estimate inertial properties. Clarkson et al. (2012) evaluated the Kinect sensor as a surface scanner using a mannequin, but found the scanning resolution to be quite low. Another approach to gain surface data is to use photogrammetry. Jensen (1978) proposed the use of stereophotogrammetry to estimate BSP parameters. In his model, the human body was divided into elliptical disks with a thickness of 20 mm, and the radii of the elliptical disks were estimated using images from the front and side. The drawback of this approach lies in the simplifying assumptions of representing body segments as the elliptical disks. However, it is possible to reconstruct the surface of a 3D object from multiple uncalibrated 2D images taken from different positions without requiring any assumptions to the geometry of the body. This principle is referred to as “structure from motion” and was initially used for producing 3D models of static objects and landscapes. Perhaps the most striking example to date is the “Building Rome in a Day” project which used images from the Flikr web site (http://www.flickr.com) to generate a 3D model of the whole city (Agarwal et al., 2009). The reconstruction of a 3D surface from multiple cameras is two-stage process. In stage one, the position, orientation and the parameters of the camera optics are estimated. This is achieved by the bundle adjustment algorithm (Triggs et al., 2000) that minimizes the error between the re-projected feature points using estimated camera pose and parameters with the actual feature points in the images. In theory, feature points could be chosen manually but this would be cumbersome and potentially not very accurate. Instead, Scale Invariant Feature Transform (SIFT) algorithms are employed which automate this process by identifying possible common points between multiple images (Lowe, 1999). Stage two uses the calibrated views to produce a dense point cloud model of the 3D object. There are a number of possible approaches to achieve this (for review see Seitz et al., 2006) but probably the most widespread current approach is patch-based multi-view stereo reconstruction (Furukawa & Ponce, 2010). This photogrammetric approach has gained wide acceptance for producing 3D models in areas such as archaeology (McCarthy, 2014) and palaeontology (Falkingham, 2012), and is even used for markerless motion capture (Sellers & Hirasaki, 2014).

The aim of this paper is to investigate whether an approach based on structure form motion photogrammetric reconstruction can provide person-specific body segment parameters, and to identify the strength and weaknesses of such an approach with regards to ease of implementation, cost-effectiveness, subject comfort and processing time. A low-cost body scanner was built using multiple cameras and the body segment parameters of six participants (four male and two female) are presented using the proposed method.

Methods

Photogrammetry relies on obtaining multiple photographs taken from different locations. These photographs can be taken with any suitable device, and for objects that do not move, the most cost-effective option is to take 50 + photographs with a single camera that is moved around the object. This has the additional advantage that a single intrinsic calibration can be used, since the camera optics can be considered identical for multiple images. However, for subjects that can move, all the photographs must be taken simultaneously so that the subject is in exactly the same position for all the images. Simultaneous photographs can be achieved in several different ways including multiple still cameras with synchronised remote controls, multiple USB web cameras, or multiple networked cameras. There is probably little to choose between these methods, but initial experimentation found that network/IP cameras provided a cost-effective solution that scaled well. The camera resolution should be as high as reasonably possible, since higher resolution images provide more information for the feature extraction algorithms and higher point density in the eventual reconstruction. This means that low-resolution cameras such as low cost web cameras and standard resolution video cameras may not be suitable.

Most applications that employ photogrammetry aim to capture surface data in great detail, with the emphasis on creating almost true-to-live 3D models and thus maximizing the point cloud density. Some applications require only the information available from the point cloud directly (such as feature point locations) and do not require a surface mesh. In fact, meshing algorithms tend to decrease the accuracy of the model (Falkingham, 2012). In applications where the reconstructed object is to be 3D-printed (Garsthagen; Hobson; Straub & Kerlin, 2014) or where volumetric data is required (such as for body segment estimations presented in this paper), a (closed) surface mesh needs to be created from the point cloud. A high-resolution mesh is commonly desired in 3D printing (e.g., for aesthetic or functional reasons), which requires a large number of photographs and sophisticated algorithms to convert the point cloud to a mesh. In this paper, we propose the use of convex hulling to generate simplified geometric outlines of the body segments. Convex hulling is robust to low-density surface point clouds (and even potential gaps in the point cloud) and can thus be implemented with ease and run automatically without requiring user input. Furthermore, being able to generate surface meshes from a low-density point cloud lowers the number of cameras required to build the 3D scanner (as opposed to needing a large number of cameras to achieve densely packed point clouds).

3D body scanner design

Photogrammetric reconstruction can work well with as few as 4 cameras (Sellers & Hirasaki, 2014) but more cameras are necessary to provide a relatively gap free reconstruction. To estimate the minimal number of cameras necessary to achieve a 360° reconstruction, we positioned a single camera on a circle of radius 1.6 m and placed a stationary skeletal dummy as a test subject in the centre. Images were taken every 5° and the point cloud reconstructions using 72, 36, 24, 18, 12 and 9 images, corresponding to angular resolutions ranging from 5° to 40°, were compared (see Fig. 1A). Acceptable reconstructions for the purpose of this paper, i.e., no loss of body segment features, were found with 18 or more cameras although using larger numbers of cameras certainly improved the point cloud density. After initial testing, the setup design was adjusted by increasing the radius of the camera placements (to increase the field of view to accommodate outstretched arms), placing the cameras above head-hight and angling the camera views downwards (as opposed to placing the cameras at the bottom or at hip-height) and using asymmetric patterns on the floor in the shared field of view of all cameras. The latter greatly aided the reconstruction reliability as the camera calibration algorithm relies on shared features.1 The network camera was implemented using Raspberry Pi (RPi) modules, type A, each equipped with an 8GB SD card and a Pi camera (http://www.raspberrypi.org). These modules run the Linux operating system (Raspbian) and provide a flexible and cost-effective 5 megapixel network camera platform. The 18 RPi modules (each with a camera) were attached to a 4.8 m diameter hexagonal frame elevated to height of 2.3 m by six support poles (see Fig. 1B). Each RPi module was provided with a USB WiFi receiver (Dynamode WL-700-RX; Dynamode, Manchester, UK) and power was provided using the standard RPi power adapter plugged into a multi-socket attached to each support pole. Four 500 W Halogen floodlights were mounted to provide additional lighting to increase the image quality.

Figure 1 Body scanner design.

(A) Point cloud reconstruction with varying number of cameras. (B) Schematic representation of the RPi scanner design.

RPi cameras can record either still images or movie files. For this application we needed to trigger all the cameras to record a single image at the same instant. This was achieved using the open source “Compound Pi” application (http://compoundpi.readthedocs.org), which uses the UDP broadcast protocol to control multiple cameras synchronously from a single server. Once the individual images have been recorded, the application provides an interface to download all the images obtained to the server in a straightforward manner. Since UDP broadcast is a one-to-many protocol, all the clients will receive the same network packet at the same time and the timing consistency for the images will be of the order of milliseconds which is adequate for a human subject who is trying to stand still. Higher precision synchronisation can be achieved using a separate synchronisation trigger but this was unnecessary in this application.

Data acquisition

Full body scans using the RPi setup were obtained from six voluntary participants. Additionally, their body weight and height was measured (Table 1). The male visible human was used as an additional data set for validation (The National Library of Medicine’s Visual Human Project (Spitzer et al., 1996)). The experimental protocol (reference number 13310) was approved by the University of Manchester ethics panel. In accordance with the experimental protocol, written consent was obtained from all participants.

Table 1 Participant mass and weight.

	P1 (m)	P2 (m)	P3 (m)	P4 (m)	P5 (f)	P6 (f)	VH (m)	
Mass (kg)	73.4	77.0	88.2	87.8	65.4	55.2	90.3	
Height (m)	1.81	1.83	1.85	1.83	1.65	1.58	1.80	
Notes.

P1–P6 Participants (m: male, f: female)

VH Male visible human

The reconstruction algorithms rely on finding matching points across multiple images so do not work well on images that contain no textural variation. We therefore experimented with using different types of clothing in the scanner, such as sports clothing, leisure clothing, and a black motion capture suit equipped with Velcro strips to aid feature detection. Clothing was either body-tight or tightened using Velcro strips if they were loose, since loose clothing would lead to an overestimation of the body volume. The participants stood in the centre of the RPi setup with their hands lifted above their head (see Fig. 2) and the 18 images were then acquired.

Figure 2 Image processing work flow.

Images from the RPI scanner are converted to 3D point clouds which are then scaled and segmented manually. Subsequently, convex hulling is used to produce a surface mesh around each body segment.

Data processing

The 3D point cloud reconstruction was initially done using open source application VisualSFM (http://ccwu.me/vsfm/) which performed adequately, but we then switched to using Agisoft PhotoScan Standard Edition v1.0.4 (http://www.agisoft.com) which proved to be much easier to install and use. Agisoft PhotoScan also achieved a better reconstruction quality with fewer holes in the point cloud and smoother surfaces.2 The parameters used in the reconstructions are reported in Supplemental Information. Agisoft PhotoScan runs identically on Windows, Mac or Linux. The full 3D reconstruction with 18 images took an average of 30 min using an 8 core 3 GHz Xeon MacPro with 12GB RAM. The actual time taken was variable depending on the image file size and the reconstruction parameters. The output of the Agisoft PhotoScan is an unscaled 3D point cloud of the participants and surrounding scenery (see Fig. 2), which requires further post-processing to calculate BSP values. First, the point cloud was scaled and oriented using CloudDigitizer (Sellers & Hirasaki, 2014), the oriented point clouds were then divided into anatomical segments using Geomagic (http://geomagic.com), and the convex hulls computed in Matlab® (http://www.mathworks.com, see Supplemental Information). The reference points for the body segmentation are listed in Supplemental Information Table S1. The body segments were all oriented into the standard anatomical pose before the volume, centre of mass and inertial tensor were calculated based on the hull shape and segment density using a custom function implemented in Matlab® (see Supplemental Information). The choice of body density is an interesting issue. Different tissues within segments have different densities and tissue composition is moderately variable between individuals. Indeed variations in density are commonly used to estimate body fat percentage (Siri, 1961; Brožek et al., 1963). MRI and CT based techniques can allow individual tissue identification and can compensate for this but surface volumetric techniques need to use an appropriate mean value. Segment specific densities are available (e.g., (Winter, 1979)) but the quoted trunk density is after subtraction of the lung volume. For a surface scan model, we need to use a lower value trunk density that incorporates the volume taken up by the air within the lungs. Therefore, for the purpose of this paper a trunk density value of 940 kg/m3 was chosen, while a uniform density of 1000 kg/m3 was assumed for all other body segments (Weinbach, 1938; Pearsall, Reid & Ross, 1994). The body mass calculated from the volume was never exactly the same as the recorded body mass, so the density values were adjusted pro-rata to produce a consistent value for total mass. (1) s=mParticipant∑mSegmHull,i.

The factor s effectively scales the body densities and is thus also applied the moments and products of inertia obtained from the convex hull segments (see Supplemental Information).

Results

Six participants were scanned using the RPi photogrammetry setup and their point cloud segmented. In order to be able to calculate the inertial properties, the point cloud needs to be converted into a closed surface mesh. To calculate the volume of an arbitrary shape defined by a surface mesh, the mesh needs to be well defined, i.e., it should be two-manifold, contain no holes in the mesh, and have coherent face orientations. The process of converting a point cloud to a well-defined mesh is known as hulling and there are several possible methods available. The simplest is the minimum convex hull where the minimum volume convex shape is derived mathematically from the point cloud (www.qhull.org). This approach has the advantage of being extremely quick and easy to perform and it is very tolerant of point clouds that may contain holes where the reconstruction algorithm has partially failed. However, it will always overestimate the volume unless the shape is convex. There are also a number of concave hulling approaches. Some are mathematically defined such as AlphaShapes (Edelsbrunner & Mücke, 1994) and Ball Pivoting (Bernardini et al., 1999) and require additional parameters defining the maximum level of permitted convexity. Others are proprietary and can require considerable manual intervention such as the built in hole-filling algorithms in Geomagic. This latter group provides the highest quality reconstructions but at the expense of considerable operator time. For this paper, we concentrated on convex hulls under the assumption that the level of concavity in individual body segments was likely to be relatively small. The relative segment mass of all participants are reported in Fig. 3 (the segmented convex hulls are shown in Fig. S1 in Supplemental Information). Figure 3 also displays average values from literature. As the participants were imaged wearing shoes, the foot volume was overestimated significantly. It is possible to adjust the value using a foot-specific scaling factor that accounts for this overestimation, although of course if the subsequent use of the BSP parameters is in experiments with participants wearing shoes then the shoe mass becomes an important part of the segment. For the purpose of this paper, a scaling factor was derived based on a single participant (P5) by comparing the convex hull volume of the foot imaged in socks versus the convex hull volume wearing shoes, and this factor (of 0.51) applied to all participants’ inertial values of the feet. The moments of inertia are shown in Fig. 4 together with average values from literature. Geometric methods also allow us to calculate the products of inertia which are otherwise simply assumed to be zero. The average products of inertia are depicted in Fig. 5 (absolute values shown only, signed values reported in Supplemental Information (Tables S2–S4). Some segments, e.g., the thigh or trunk, have products of inertia that are of a similar order of magnitude as their moments of inertia, which is indicative of a noticeable difference between the inertial principal axes and the anatomical principal axes. However, the majority of the products of inertia are significantly smaller than the moments of inertia (of the same segment) by one to two orders of magnitude. Figure 6 contains the relative centre of mass in the longitudinal segment direction, i.e., along the z-axis with the exception of the foot whose longitudinal axis corresponds to the x-axis (see Fig. 2). Figure 7 shows the shift of CoM from the longitudinal axis in the transverse plane (x–y plane). The CoM values in literature assume a zero shift from the principal anatomical (longitudinal) axis. The shift values we found with our geometric method are generally unequal to zero, but they have be to viewed with caution as the placement of the reference anatomical axis itself has uncertainties associated with it. The numerical values presented in Figs. 3–7 and the segment lengths are reported in Supplemental Information (Tables S2–S13).

Figure 3 Segment mass (as % of body mass).

P, Average value of all six participants (error bars show standard deviation). Foot mass adjusted by a factor of 0.51 to compensate for volume overestimation due to wearing shoes. Z(m), Male average values reported by Zatsiorsky; Z(f), Female average values reported by Zatsiorsky (Leva, 1996; Zatsiorsky, 2002); D(m), Male average values by Dempster (via Zatsiorsky) (Dempster, 1955; Zatsiorsky, 2002).

Figure 4 Moment of inertia in (104 kg m2)

P, Average value of all six participants (error bars show standard deviation). Foot moment of inertia adjusted by a factor of 0.51 to compensate for volume overestimation due to wearing shoes. Z(m), Male average values reported by Zatsiorsky; Z(f), Female average values reported by Zatsiorsky (Leva, 1996; Zatsiorsky, 2002). The definition of the coordinate system is shown in Fig. 2.

Figure 5 Absolute values of products of inertia in (104 kg m2).

The absolute values of Ixy, Ixz and Iyz are shown together with a positive error bar (negative error bar is symmetrical) equal to one standard deviation. The signed values are reported in Supplemental Information (Tables S2–S4). The Ixy value of the hand is smaller than 103 kg m2 and is not displayed. Foot products of inertia adjusted by a factor of 0.51 to compensate for volume overestimation due to wearing shoes.

Figure 6 Centre of mass along the longitudinal axis.

P, Average value of all six participants (error bars show standard deviation). Z(m, male; f, female): Average values by Zatsiorsky, adjusted by de Leva. The CoM is given as % of the segment length. The definition of the segments and reference points are given in Supplemental Information Table S1 - Exceptions: * Foot of participants: Heel and toe end point of participant’s shoes instead of foot. ** Forearm and Upper Arm of Z: Elbow reference point is the elbow joint centre instead of the Olecranon (Leva, 1996; Zatsiorsky, 2002).

Figure 7 CoM shift from the anatomical longitudinal axis in the transverse (x–y) plane.

Average values of all six participants are shown (error bars show standard deviation). Due to mirror-symmetry, the y-values of the segments on the left- and right-hand side have opposite signs. To calculate the average, the sign of the segments on the left-hand side was inverted. The CoM is given as % of the segment length. The data of the foot is not included due to the participants wearing shoes.

To estimate the effect of the convex hull approximation on the mass estimation versus the original body segment shape, the volumes of a high resolution 3D body scan and of their convex hull approximation were calculated and compared. A detailed surface mesh was obtained from the National Library of Medicine’s Visible Human Project (Spitzer et al., 1996) by isosurfacing the optical slices using the VTK toolkit (http://www.vtk.org) and cleaning up the resultant mesh using Geomagic. The surface mesh of the 3D body scan was separated into body segments and the volume calculated following the same methodology as used for the point cloud data. A convex hull was applied to each body segment and the volume calculated again (see Fig. 8). The volume overestimations for each body segment (averaged between left and right) are shown Fig. 9 (column CH). Several body segments showed a large relative volume overestimation (using 10% error as a cutoff, although the choice would depend on the required accuracy): foot (26%), shank (31%), hand (47%) and forearm (16%). This is due to the relatively strong curvatures in these segments. To minimize the effect, these body segments were subdivided (see Fig. 10) and the convex hulls recalculated. The results of the divided segments are also shown in Fig. 9 (column CHD), and the decrease in volume overestimation is apparent. The volume overestimation of the subdivided foot (11%), shank (11%) and forearm (5%) are at a similar level to the other body segments and would probably be acceptable in many cases. The hands show the largest relative mass overestimation still (25%), which is due to its curved position and slightly open fingers. The convex hull error of the hand is, however, expected to be significantly smaller if the hand is imaged while being held in a straight position with no gaps between the digits.

Figure 8 Visible human surface mesh.

(A) High-resolution surface mesh. (B) Convex hull mesh.

Figure 9 Segment volume overestimation of the hulled mesh versus the high-resolution surface mesh of the visible human.

Data shown as the relative difference of the hull with respect to the original mesh. CH, Convex hull of body segment; CHD, Convex hull of divided body segments (only segments indicated with an * were subdivided, see Fig. 10).

Figure 10 Subdivision of the body segments with large curvature.

The first row (S) shows the high-resolution surface mesh, the second row (CH) the convex hull of the whole body segment, and the bottom row (CHD) the convex hulls of the subdivided body segments.

Figure 11 contains the relative mass estimations of the original surface mesh, the convex hulls with and without subdivision, and the average and regression model values found in literature. With a BMI value of almost 28, the male visible human is not well represented by the average or regression model values found in literature, where the majority of the studies involve relatively athletic people (BMI average of around 24) or obese individuals (BMI over 30). The convex hulls of the subdivided segments (CHD in Fig. 11) give the closest approximation to the original mesh and, with the exception of the hands, are within a relative error of less than 5%. The relative error of the convex hull of the whole segments (CH in Fig. 11) is larger but closer to the original mesh than average and regression values given in literature. The moments of inertia are overestimated as well as they are a product of the mass of the segment. Their overestimation follows the same trend as the mass overestimation, i.e., the largest overestimation occurs for the hands, followed by the shanks and feet (see Fig. S2 in Supplemental Information), and the subdivided segments produce more accurate values with an average relative error below 10%.

Figure 11 Male visible human segment mass (as % of body mass) of the high-resolution mesh, convex hull, regression model and average values.

S, High-resolution surface mesh; CH, Convex Hull of whole body segments; CHD, Convex Hull with subdivided body segments (only segments indicated with an * were subdivided as shown in Fig. 10); ZR, Values predicted using Zatsiosrky’s linear regression model (using weight and height); Z, Male average values reported by Zatsiorsky; D, Male average values reported by Dempster (Dempster, 1955; Leva, 1996; Zatsiorsky, 2002).

Discussion

We can see from the results that the proposed methodology produces values that are very similar to those derived using regression equations. There are no consistent problems, although it is clearly important that the hand is held in a suitable flat position but with fingers adducted so that the hulling can provide an accurate volume estimation. We would expect that the photogrammetric process will work as well as any of the published geometrical approaches (Hanavan, 1964; Hatze, 1980) since it is simply an automated process for achieving the same outcome. The procedure is currently moderately time consuming in total, but the interaction time with the participant is extremely short and involves no contact, which can be very beneficial for certain experimental protocols or with specific vulnerable participants. Since most of the time is spent post-processing the data, we expect that this post-processing could be streamlined considerably by writing dedicated software rather than the current requirement of passing the data through multiple software packages.

In general, regression equations work well for applicable populations and are probably more suitable if body mass distribution is not a major focal point of the research, particularly given that in some cases it can be shown that experimental outcomes are not especially sensitive to the BSP parameters chosen (Yokoi et al., 1998). The values generated in our sample are relatively close to those generated by using regression equations but BSP values are highly variable between individuals and current regression equations are only suitable for a very limited range of body shapes. This is particularly the case when we are dealing with non-standard groupings such as children, the elderly or people with particularly high or low BMI values.

However there are some specific issues with this technique that could to be improved for a more streamlined and potentially more accurate workflow (see Fig. 12, which summarises the steps involved in estimating body segment parameters using photogrammetry).

Figure 12 Methodology to estimate subject-specific body segment parameters using photogrammetry.

(A) Photogrammetry; (B) Body segmentation; (C) Segment hulling; (D) Inertial parameter estimation.

Convex hulling of the point cloud is a robust and fast way to produce surface meshes. The fact that it systematically overestimates the volume of concave features can be improved by subdividing body segments into smaller parts and the decision then becomes what level of subdivision is appropriate for an acceptable level of accuracy (see Fig. 12C). For example, with only one subdivision of the shank and forearm the relative error of their volume overestimation was reduced by a factor of three, and the end result was within 10% of the true value which is probably sufficient in most cases, especially given the level of uncertainty in other parameters such as segment specific density. It is important to note that the scaling factor used in our method significantly minimises the segment mass estimation errors introduced by the volume overestimation. In fact, if all hulled segment masses (i.e., the product of segment volume and segment-specific density, see Fig. 12D) were overestimated by 10%, the final body segment mass would be calculated correctly due to the scaling factor applied to each segment. Therefore, using a pro-rata scaling factor performs best when the relative errors of the volume estimation of each segment are within a small range of each other.

The adoption of one of the concave hulling techniques is likely to lead to a similar level of improvement again with a minimum (but not zero) level of additional work. The level of subdivision required not only depends on the body segment, but also the population studied so it may well be appropriate that the segmentation level is adjusted according to the type of study and its sensitivity to inaccuracies in the BSP (i.e., multiple segment subdivisions increase accuracy of volume estimation). In this work, a uniform scaling factor and constant body density (apart from the trunk) was assumed. It is well known that the density varies among body segments as well as among populations due to different percentages of fat and muscle tissue (Drillis, Contini & Bluestein, 1964; Durnin & Womersley, 1974; Zatsiorsky, 2002). Thus, using segment and population specific densities (and scaling factors) may improve the accuracy of the presented methodology if such values are available or derived. Similarly, important contributions to segmental mass distribution such as the presence of the lungs within the torso can be explicitly modelled, which may lead to small but important shifts in the centre of mass (Bates et al., 2010).

In terms of technology, the current arrangement of using 18 Raspberry Pi cameras is reasonably straightforward and relatively inexpensive. It requires no calibration before use, and the process of moving the subject into the target area is extremely quick. However, it does take up a great deal of room in the laboratory, and the current software is reliant on clothing contrast for the reconstructions, which limits the flexibility of the technique. This could be improved by projecting a structured light pattern onto the subject so that areas with minimal contrast can be reconstructed accurately (Casey, Hassebrook & Lau, 2008). Our results show that 18 cameras is currently the minimum needed for full body reconstruction, and a system with 36 or more cameras would produce better point cloud reconstruction results by minimizing areas of potential occlusions (such as between the legs or between the arms and trunk) and increasing the point cloud density. To what degree a more densely packed point cloud would significantly improve the accuracy of the estimated inertial parameters based on convex hulls would be an interesting aspect to investigate further. We would expect a denser point cloud to facilitate the use of more complex meshing methods instead of convex hulling.

One future use of this technology is clearly the use of such systems and algorithms for complete motion capture (Sellers & Hirasaki, 2014). The limitation currently is that these cameras would need to be closely synchronised, and whilst the proposed system is adequate for producing a single still image, it is currently not able to adequately synchronise video. In addition, the video resolution is much lower and this makes the reconstruction more difficult. However, we predict that markerless, multiple video camera structure from motion systems will become a much more common mainstream tool for experimental motion capture in the near future. Ideally, we could imagine that such a system would both do the motion capture and also the body segment parameter reconstruction, since much of the computational technology would be shared.

Conclusion

A methodology based on structure form motion photogrammetric reconstruction has been presented that provides subject-specific body segment parameters. The method relies on the surface depth information extracted from multiple photographs of a participant, taken simultaneously from multiple different view points. The brief interaction time with the participants (taking all required photos simultaneously, and measuring the height and weight only) makes this a promising method in studies with vulnerable subjects or where cost or ethical constraints do not allow the use of other imaging methods such as CT or MRI scans. Unlike regression models that are valid only for a small population sample, we expect the proposed methodology to be able to perform equally well for a wide range of population samples. The post-processing time is lengthy compared to using regression models or average values from literature but not compared to processing MRI or CT data. The 3D scanner presented in this paper was able to produce a sufficient 3D data points to estimate body segment volumes with only 18 RPi cameras, which kept the hardware cost to a minimum. Depending on the accuracy required for the project, we would expect both more cameras and higher resolution cameras to improve the robustness of the 3D point cloud reconstruction.

While the results presented in this work were derived using commercial software such as AgiSoft, Geomagic and Matlab®, we were able to to achieve similar results using open-source software only (such as VisualFMS (http://ccwu.me/vsfm/) for calculating 3D point clouds and MeshLab (http://meshlab.sourceforge.net/) for point cloud segmentation, hulling and BSP calculation). This makes our proposed methodology, in combination with the low hardware costs, particularly promising for small-budget projects.

Supplemental Information

Supplemental Information Supporting Information

Click here for additional data file.

The authors would like to thank Dave Jones for the development of the Compound Pi programme and his generous help with the network setup of the Raspberry Pi scanner.

Additional Information and Declarations

Competing Interests

Author Contributions

Human Ethics

1 Without the patterns on the floor, the camera calibration relied on shared features found on the subject, whereas the patterned floor provided a large (or even completely sufficient) number of features to run the camera calibration algorithm.

2 This is based on the comparison of the best reconstruction result achieved with each software after testing an extensive, but not complete, combination of reconstruction parameters.

The authors declare there are no competing interests.

Kathrin E. Peyer conceived and designed the experiments, performed the experiments, analyzed the data, wrote the paper, prepared figures and/or tables, reviewed drafts of the paper.

Mark Morris conceived and designed the experiments, performed the experiments, analyzed the data, reviewed drafts of the paper.

William I. Sellers conceived and designed the experiments, wrote the paper, reviewed drafts of the paper.

The following information was supplied relating to ethical approvals (i.e., approving body and any reference numbers):

The experimental protocol (reference number 13310) was approved by the University of Manchester ethics panel.

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
