# Peer review of "Subject-specific body segment parameter estimation using 3D photogrammetry with multiple cameras"

_PeerJ, doi:10.7717/peerj.831_

## Round 0.1 · original submission · Minor Revisions

You have two reviews. One suggesting major revisions, one minor. My opinion is that it falls in between. But I like the paper and certainly find it publishable. To make the paper more broad and inclusive, some elaboration of background and less successful attempt to accomplish similar goals might be useful. You also might elaborate the broader impacts and intellectual merit in the conclusions.

·

Basic reporting

all fine

Experimental design

overall, all fine. However, I have marked a few places in the MS where the authors could, maybe even should, go into more details (maybe in the form of an appendix) so that using their method becomes easier for others.

Validity of the findings

I didn't check every individual number. That said, overall everything looks fine. Also, the authors are appropriately cautious and detail clearly what their results are versus conclusions and speculation.

I would like to see two or three extreme data points added - a very obese person or a child. Not a must, but it would go a long way to distinguish this study further from typical regression-based approaches, as it would highlight the strengths very well (assuming it works).

Additional comments

There are a few spelling etc. issues, please fix.
Otherwise, a very good paper.
As I commented in the MS PDF, there are a few things I would like to see added or detailed better - not a must, though!

Sorry I took the full ten days for this! I had other reviews with short deadlines in parallel, otherwise this could have been done in one day.

Reviewer 2 ·

Basic reporting

Specific applications and purposes for the results generated by the proposed technique do not appear adequate to fully understand how they will be used.

Experimental design

All photos were taken at an elevation of 2.3 meters. Since photogrammetry is a line-of-site technique, would not the addition of lower angle images be important in producing a more complete and accurate model?

“…using larger numbers of cameras certainly improved the reconstruction quality.” To what degree? Would better reconstruction quality improve the parameter estimations? Can some level of comparative data be provided if more cameras were used?

The authors’ repeated acknowledgement of the “overestimations” caused by the test subjects’ clothing and physical pose (i.e., hand and finger position) raises the question as to why were more appropriate measures not used during testing (e.g., wearing of swim or workout attire and no shoes)? Are there alternatives that can be used to accommodate certain cultural or social requirements that might be more accurate than those implemented in your test?

What is presented appears to have been a single procedure conducted one time (e.g., number of cameras, their elevation, the clothing and position of subjects), If other alternative methods were attempted, what were the results and how do they compare to what you have presented?

Validity of the findings

Are the “average values presented in the literature” (lines 176, 182, 302, Figures 3-7) accepted within the discipline as accurate or valid? Is there a range of variability of the body segment parameters that is acceptable? If so, do the results from this proposed technique fall within appropriate ranges?

Some of the statements seem generalized, and specific requirements or objectives are not provided. Is the cost and time required of the technique more important than the accuracy of the results?

Why is there no reference to related work being done in the private/corporate sector?

---

## Round 0.2 · accepted · Accept

Well done. thank you. great paper